# Unimolecularly thick monosheets of vinyl polymers fabricated in metal–organic frameworks

Nobuhiko Hosono [1,2], Shuto Mochizuki[1,3], Yuki Hayashi[1] & Takashi Uemura [1,2,4 ✉]

Polymers with two-dimensional (2D) network topologies are currently gaining significant attention due to their unique properties that originate from their regulated conformations. However, in contrast to conventional 1D- and 3D-networked macromolecules, the synthesis of such 2D networks provides challenges for polymer chemists because of the nature of the networking polymerisation reaction, which occurs in a spatially random fashion when conventional solution-phase synthesis is performed. Here we report a versatile synthesis of polymeric monosheets with unimolecularly thick networking architectures by exploiting the 2D nanospaces of metal–organic frameworks (MOFs) as reaction templates. Crosslinking radical polymerisation in the 2D nanospaces of pillared-layer-type MOFs affords monosheets of typical vinyl polymers and can be carried out on the gram scale. Remarkably, the prepared polymer monosheets are highly soluble in organic solvents and show atypical thermal and rheological properties that result from their 2D-regulated conformations that cannot be adopted by their 1D or 3D analogues.

[1] Department of Advanced Materials Science, Graduate School of Frontier Sciences, The University of Tokyo, 5-1-5 Kashiwanoha, Kashiwa, Chiba 277-8561, Japan. [2] Department of Applied Chemistry, Graduate School of Engineering, The University of Tokyo, 7-3-1 Hongo, Bunkyo-ku, Tokyo 113-8656, Japan. [3] Department of Synthetic Chemistry and Biological Chemistry, Graduate School of Engineering, Kyoto University, Katsura, Nishikyo-ku, Kyoto 615-8510, Japan. [4] CREST, Japan Science and Technology Agency (JST), 4-1-8 Honcho, Kawaguchi, Saitama 332-0012, Japan. ✉email: t-uemura@k.u-tokyo.ac.jp

Polymers with two-dimensional (2D) network topologies have emerged as an intriguing class of macromolecule[1–4]. Their 2D molecular shapes give rise to many unique electronic[5,6], magnetic[7], catalytic[8], and mechanical properties[9] that are not manifested in the three-dimensional (3D) bulk state of the parent polymers. So far, however, the successful syntheses of such macromolecules are limited to highly specific systems that exploit topochemical reactions of crystalline monomers[10–12], molecular tessellations and polymerisations at surfaces/interfaces[13–18], and the exfoliations of layered crystalline frameworks[19–24]. Although such top-down approaches are promising for producing 2D materials with better productivities and yields, these examples essentially require highly symmetrical multivalent monomers to define the dimensionalities of products in advance, which precludes versatility. Vinyl polymers, which are used for many value-added plastic raw materials and account for the largest need of the current petrochemical industry, are generally synthesised by solution-phase polymerisation reactions. The solution-phase reactions of vinyl monomers in the presence of multivalent agents result in random branching and crosslinking that generally lead to hyperbranched structures with 3D expanded conformations or otherwise less-soluble networked bulk materials. By carefully choosing the reaction media, thin-layer networks of vinyl polymers have been synthesised by exploiting the crosslinking polymerisation of monomers that are placed at surfaces/interfaces[25,26], in self-assembled layers[27–31], and in clay-monomer clathrates[32]; however, achieving accuracy and scalability in these reaction systems remain formidable challenges. For non-crystalline low-symmetric vinyl monomers, dimensionally programmed polymerisation is required to render them into unimolecularly thick 2D networks by regulating the conformational dimensionality of the growing radical chains.

Metal–organic frameworks (MOFs) are synthetic microporous materials prepared by the self-assembly of metal ions and bridging organic ligands and have attracted attention as potential platforms for host–guest chemistry, heterogeneous catalysis, and molecular adsorption/separation[33–37]. One of the distinctive feature of MOFs, among other microporous materials, is its high degree of designability where, for example, pore size and shape, surface functionality, and channel dimensionality can be reasonably tuned by changing the building components[38]. MOFs have also been used as host materials for polymerisation because they facilitate the regulation of the primary and higher order structure of the polymer[39–44]. In particular, the pore dimensionality of a MOF can be transcribed into the generated polymer, as demonstrated recently by MOF-templated polymerisation[39–41].

Herein, we describe a universal and gram-scale synthesis of ultimately thin vinyl polymer networks with controlled thicknesses by exploiting confined radical polymerisation reactions in pillared-layer-type MOFs (Fig. 1), which affords unimolecular monosheets of various vinyl polymers. The MOF nanospaces which have crystallographically defined structures allow us the highest precision of thickness control without sacrificing monomer versatility for 2D material syntheses. We use the pillared-layer-type [Ni(Hbtc)(bpy)]$_n$ MOF (1) (btc = 1,3,5-benzenetricarboxylate, bpy = 4,4′-bipyridyl)[45,46], which has a layered structure with 2D interstices, as the nano-container for the crosslinking radical polymerisations of mono- and di-functional vinyl monomers (Figs. 1 and 2). Decomposition of the host MOF after polymerisation leaves polymer 2D sheets that are very soluble in appropriate solvents and display monosheet structures when deposited on a substrate. The fundamental properties of a macromolecule, such as its thermal and mechanical properties, originate not only from its molecular structure and molecular weight, but also its molecular shape. The 2D polymer sheets fabricated by our approach using MOFs exhibit quite unique thermal and mechanical properties in the bulk state; they have lower glass transition temperatures ($T_g$) compared to those of their linear analogues, while crosslinking reactions generally result in higher $T_g$ values[47]. In addition, the polymer sheets exhibit atypical rheological behaviour as evidenced by the complete absence of a rubbery plateau during dynamic mechanical analysis (DMA), which is plausibly due to suppressed entanglements and fast relaxation processes ascribable to the distinctive 2D network topology of its molecular shape.

## Results

**Preparing the polymer monosheets.** The pillared-layer [Ni(Hbtc)(bpy)]$_n$ MOF (1) was used as the host for crosslinking polymerisation (Fig. 2)[45,46]. Since 1 has nanospaces that are comparable in size to conventional vinyl monomers, the chain-growth reaction is only allowed in the *ab* planes of the 2D spaces of 1, thereby providing monosheets of the crosslinked polymer network. Styrene (St: molecular size = 6.8 × 4.4 Å) and ethylene glycol dimethacrylate (EDMA) were incorporated into the host as the guest monomer and crosslinker, respectively. The [Ni(Hbtc)]$_n$ layers are linked by pillar bpy ligands to form the pillared-layer structure, in which each layer is separated from the next by ~0.8 nm (Fig. 2). Interlayer communication among the guest monomers is not possible because of the narrow aperture of the [Ni(Hbtc)]$_n$ layer in 1, which is ~3 Å in diameter; this impedes interlayer polymerisation reactions and leads to anisotropic networking in the 2D interstices of the host MOF. Molecular dynamics (MD) simulations of St placed in the interstices of 1 support the view that monomer hopping to adjacent 2D spaces cannot occur on a reasonable timescale at a reasonable temperature (Fig. 2c, d and Supplementary Fig. 1), which motivated us to carry out crosslinking reactions in 1 that would give unimolecularly thick monosheets of polymer networks.

The crosslinking radical polymerisation of St in 1 was performed as follows (Fig. 1). A weighed amount of activated 1 was immersed in a mixture of St, EDMA, and a radical initiator (azobisisobutyronitrile; AIBN) in order to incorporate these guests into 1 (see Methods). Excess St on the outsides of the MOF crystals were then removed at room temperature under reduced pressure (0.2 kPa) to give the host–monomer composite (1⊃St). The radical copolymerisation of St and EDMA was then induced by heating the composite at 70 °C for 48 h under nitrogen to give the host–crosslinked polystyrene (PSt) composite (1⊃PSt), after which a HCl/MeOH solution was added to the composite to digest the MOF and leave the crosslinked PSt (PSt-1) as a white powder.

The conversions of St and EDMA were individually determined to be 66% and 100%, respectively, by $^1$H NMR spectroscopy of the composite before and after the reaction, where the composite was digested by a DCl/DMSO-$d_6$ mixture in situ to liberate unreacted monomers (see Methods). The degree of crosslinking (molar ratio of EDMA to St) of the resultant PSt-1 was calculated to be ~1% (Supplementary Fig. 2). The crystal structure of the host MOF was retained even after polymerisation, as confirmed by X-ray powder diffractometry (XRPD) of the composite before and after polymerisation (Fig. 3a). In addition, scanning electron microscopy (SEM) revealed that the original hexagonal shape of crystalline 1 was maintained during the entire reaction process (Supplementary Fig. 3). After polymerisation, the MOF template was removed by treatment with a HCl/MeOH solution to leave PSt-1. The complete removal of the MOF host was confirmed by XRPD where the total absence of the XRPD pattern of 1 (Fig. 3a). This was also supported by SEM-energy-dispersive X-ray spectroscopy and X-ray fluorescence analysis (Supplementary Fig. 4). The isolated particles of PSt-1 preserved the original hexagonal shape of the parent crystal of 1 even after the removal of the crystalline host

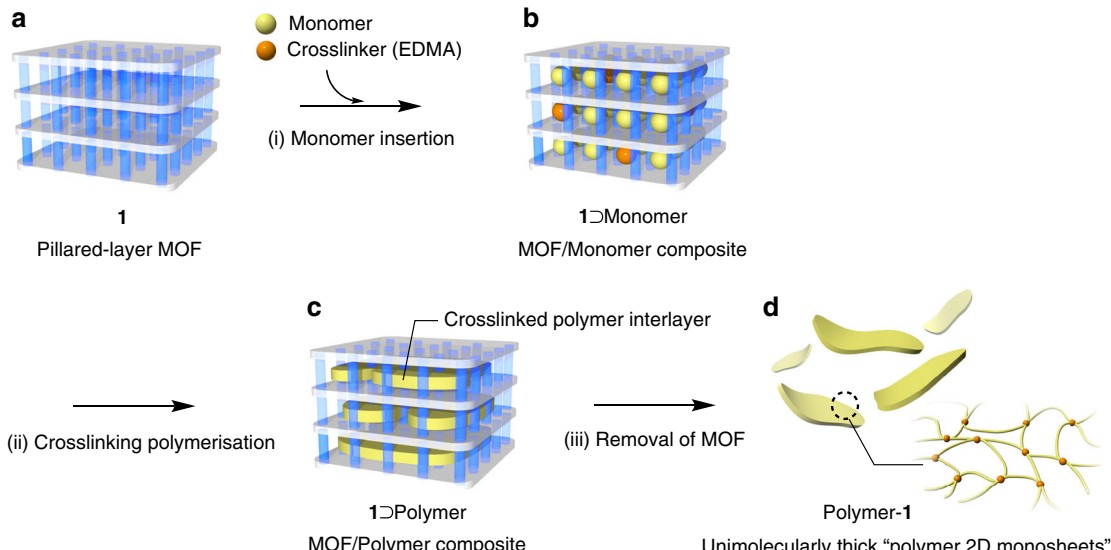

**Fig. 1 Synthesis of polymer 2D monosheets using a pillared-layer MOF. a–d** Schematically depicting the fabrication of ultimately thin polymeric sheets by crosslinking polymerisation in a MOF with 2D nanospaces. **a** Pillared-layer MOF, **1**. **b 1**⊃Monomer composite. **c 1**⊃Polymer composite. **d** Polymer-**1** with a unimolecularly thick polymer networking architecture.

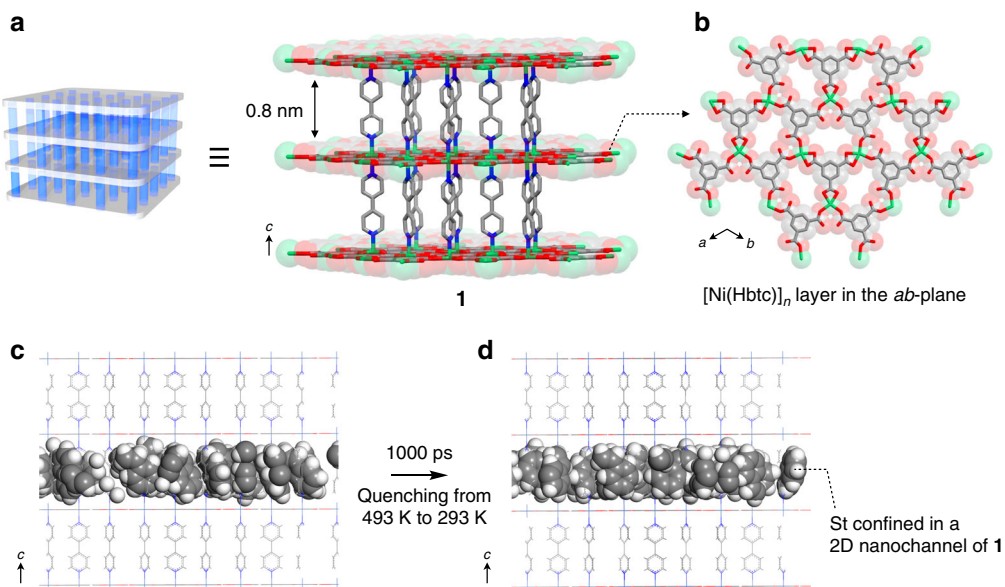

**Fig. 2 MOF structures. a** Crystal structure of **1**. **b** The structure of the [Ni(Hbtc)]$_n$ layer. Atoms: Ni (green), O (red), C (grey), and N (blue). H atoms are omitted for clarity. Disordered bipyridine moieties have been omitted for clarity in **a**. An MD simulation snapshot of **1**⊃St **c** before and **d** after quenching the dynamics from 493 to 293 K over 1000 ps under NVT condition, where the St monomer units are confined within a single interstice of **1**. The MD simulations reveal that the St monomers do not leap across [Ni(Hbtc)]$_n$ layers into neighbouring interstices (Supplementary Fig. 1).

(Supplementary Fig. 3). These observations suggest that polymerisation proceeded only inside the host MOF. This MOF-templated polymerisation reaction is tolerant to large-scale operation and can afford grams of polymer monosheets simply through the use of the corresponding amount of the MOF host. Such reaction scalability is highly advantageous when compared to that of conventional thin-layer polymerisation techniques that use surface/interface-deposited monomers[13–15,25,26], and will accelerate 2D material sciences and characterisation that have been impeded to date due to a lack of production and supply of materials.

**Structure of the polymer monosheet.** Despite the crosslinked structure, PSt-**1** is very soluble in solvents that typically dissolve

conventional PSt (e.g., chloroform, toluene, and 1,2-dioxane); in solution it exhibited a Tyndal effect, which is indicative of a colloidal dispersion of PSt-**1** in the solvent (Fig. 3b). The size distribution of PSt-**1** in the solution phase was determined by dynamic light scattering (DLS) experiments (Fig. 3c); PSt-**1** in chloroform showed a monomodal DLS peak that corresponds to a diameter of 98 nm. In order to determine molecular weight information for PSt-**1**, we subjected both PSt-**1** and its linear analogue to static light scattering (SLS) analysis. Linear PSt, hereafter referred to as PSt-L, was prepared by conventional solution-phase free-radical polymerisation (Supplementary Methods), which resulted in a polymer with $M_w$ = 238,000 g mol$^{-1}$ and $M_w/M_n$ = 2.0 as determined by size-exclusion chromatography (SEC) calibrated using PSt standards. The SLS results for PSt-L

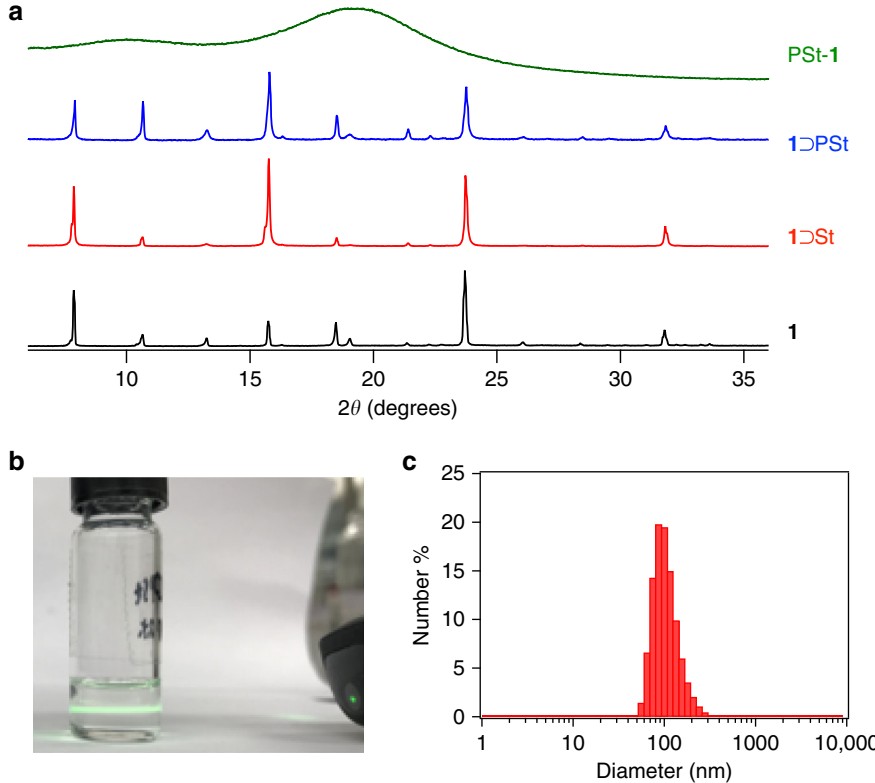

**Fig. 3 Synthesis of 1⊃PSt and PSt-1. a** XRPD patterns of **1**, **1**⊃St, **1**⊃PSt, and isolated PSt-**1**. **b** The Tyndall effect observed by passing a laser beam through the chloroform dispersion of PSt-**1**. **c** Size distribution of PSt-**1** measured by DLS in chloroform at 25 °C.

provided an $M_w$ of 233,000 g mol$^{-1}$ with an $R_g$ (radius of gyration) of 35.1 nm. The SLS-based $M_w$ is in good agreement with that determined by SEC. On the other hand, the SLS data for PSt-**1** provided an $M_w$ of 298,000 g mol$^{-1}$ and an $R_g$ of 59.3 nm (Supplementary Fig. 5); hence PSt-**1** exhibits a similar absolute $M_w$ to its linear analogue, but a 1.7-fold larger $R_g$. This observation presumably indicates that PSt-**1** possesses an anisotropically expanded conformation compared to a linear analogue with a comparable $M_w$ that theoretically adopts a random coil conformation with a spherical projection geometry.

As a necessary result of the narrow interlayer distance in **1** (~0.8 nm), chain growth and crosslinking reactions in **1** proceed in 2D directions in its interstices, resulting in the formation of a crosslinked polymer network that is only unimolecularly thick. The actual morphology of an individual PSt-**1** sheet was imaged by atomic force microscopy (AFM) (Fig. 4). Figure 4a, c shows the AFM images of PSt-**1** deposited on a highly oriented pyrolytic graphite (HOPG) substrate by spin casting from a dilute chloroform solution (0.1 μg mL$^{-1}$), and clearly reveals that PSt-**1** has a uniformly thick 2D sheet structure. Height profile analysis suggests that each PSt-**1** sheet is ~0.7 nm thick, which is in good agreement with the cross-sectional diameter of a single PSt chain (i.e., the size of the longer axis of St). The network structure of individual PSt-**1** monosheet could be roughly estimated by numerical calculation based on NMR data and the crosslinking density. Each PSt-**1** monosheet has ~30 crosslinking points interconnected with PSt chains with the molecular weight of ~5 kg mol$^{-1}$, forming the mesh-like 2D networking structure (Supplementary Fig. 6).

The high designability of the MOF host structure facilitates highly precise control of the thickness of the polymeric nanosheet. Use of a longer pillar ligand, namely 1,4-di (4-pyridyl)benzene) (dpb), gave an analogous MOF, namely [Ni (Hbtc)(dpb)]$_n$ (**2**), which has a larger interlayer distance of 1.2 nm

(Supplementary Fig. 7). Polymerisation of St in **2** also gave polymer 2D sheets (i.e., PSt-**2**) using the same reaction protocol to that involving **1**. AFM images of PSt-**2** deposited on the HOPG substrate show a 2D sheet structure (Supplementary Fig. 8) that is thicker than that of PSt-**1**. The thickness PSt-**2** is comparable to the interlayer distance in **2**, suggesting that the thickness of the crosslinked polymer sheet can be successfully controlled by tuning the host MOF structure.

In contrast to the 2D-expanded morphology observed on the HOPG substrate, PSt-**1** behaves differently on a mica substrate, where it adopts a much thicker spherical shape (Fig. 4e, f). This outcome can be explained with reference to the polarity of the polymer and its interactions with the substrate. PSt is a low-polarity polymer with a high affinity for non-polar substrates, such as HOPG. On the other hand, interactions between PSt and the mica substrate are expected to be poorer than those involving HOPG because a freshly cleaved mica surface is highly polar due to exposed oxygen atoms. Stronger binding at the polymer/ substrate interface gives rise to an expanded polymer sheet conformation upon deposition, with better contact and surface coverage. However, this is not the case for the lower affinity polymer/mica system, resulting in the folded, and rather aggregated particle morphology observed on the mica substrate. This dynamic change in particle morphology is possibly the consequence of the distinct monolayer configuration of the PSt network. The unimolecularly thick polymer monosheets have much higher degrees of freedom and, as a consequence, can adopt multiple possible conformations, while conventional crosslinked nanoparticles (e.g., microgels) can only adopt spherical conformations, as generally observed and reported in the literature[48,49].

This MOF-templated crosslinking polymerisation technique facilitates the synthesis of poly(methyl methacrylate) (PMMA) monosheets in the same manner as observed for PSt. The

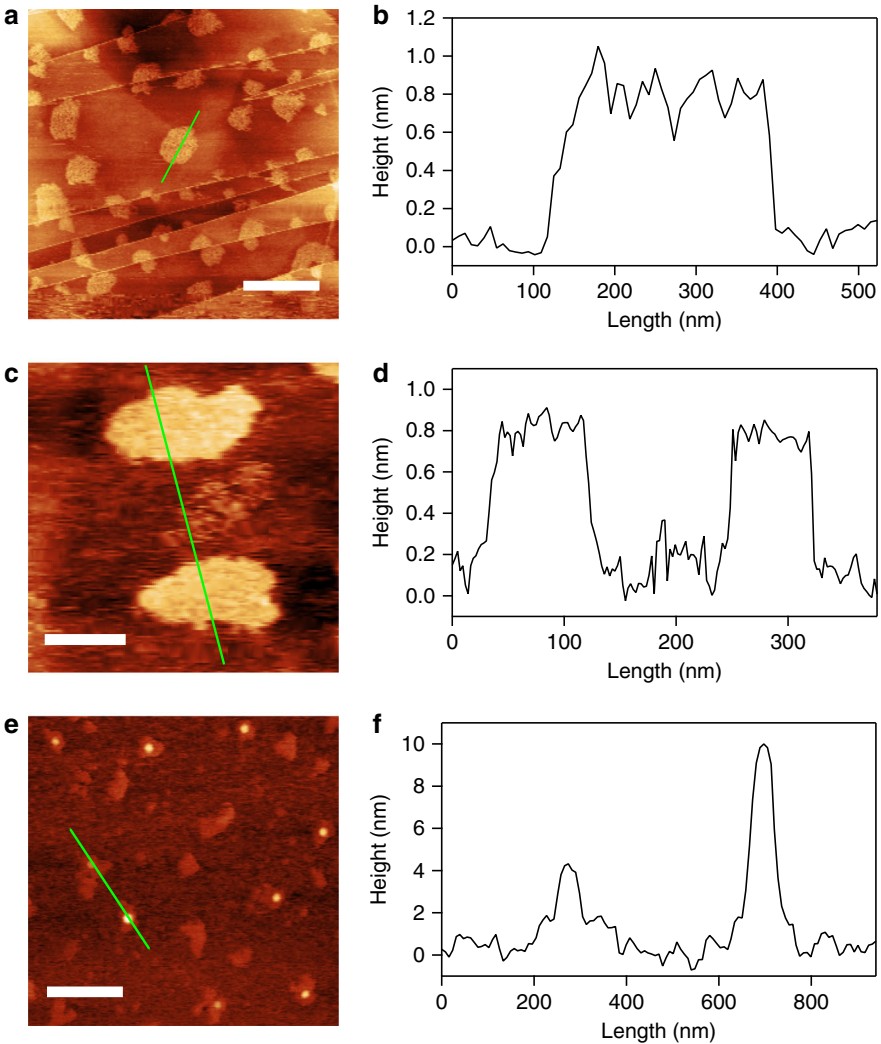

**Fig. 4 AFM images of PSt-1 deposited on HOPG or mica substrates. a** AFM topographic image and **b** height profile of PSt-**1** deposited on an HOPG substrate. Scale bar: 500 nm. **c** A magnified AFM topographic image and **d** height profile of independent sheets of PSt-**1** deposited on an HOPG substrate. Scale bar: 100 nm. **e** AFM topographic image and **f** height profile of PSt-**1** deposited on a mica substrate. Scale bar: 500 nm. Each height profile follows the green line in the corresponding AFM image. All materials were spin-cast on the given substrate from a 0.1 μg mL$^{-1}$ chloroform solution.

molecular size of methyl methacrylate (MMA: 5.9 × 4.1 Å) suggests that MMA monomers will be successfully confined in **1** and not undergo undesired interlayer crosslinking (Supplementary Fig. 9). The crosslinking polymerisations of MMA was performed in **1** in the presence of EDMA and AIBN, to afford the respective polymer networks, namely PMMA-**1**. The morphologies of PMMA-**1** were analysed by AFM; the PMMA networks exhibit flat sheet structures with only single-chain thicknesses (Supplementary Fig. 10), as expected. Interestingly, the sheet-like morphology of PMMA-**1** was observed only when deposited on the mica substrate, while large aggregated particles were observed when HOPG was used as the substrate (Supplementary Fig. 10). These experimental observations are reflective of the high polarity of PMMA which has polar carbonyl-group-containing side chains. These observations are in stark contrast to that of apolar PSt-**1**, which highlights that polymer–substrate affinity plays a crucial role in controlling the morphology of a flexible polymer monosheet.

It should be noted that the size of polymer monosheets obtained was generally much smaller (~100 nm) than the size of host MOF crystals (~50 μm). This is typically observed in the confined polymerisation system using MOFs[42,44]. AIBN loaded in the MOF generates free radicals to start polymerisation at random locations in the crystals. Volume shrinkage during polymerisation (e.g., volume of dimer < volume of two monomers) results in the formation of many polymer domains isolated in the MOF crystals. In addition, diffusion of the polymer domains is highly restricted because of the confined feature of this reaction system. This situation leads to depletion of monomers, which ends up with smaller size of product compared to the size of host MOF crystals. As also evidenced by the AFM-based analysis of PSt-**1**, which showed size distribution of the monosheets (Supplementary Fig. 11), the size control is the next challenge for this system.

**Polymer monosheet properties**. The thermal properties of PSt-**1** are expected to differ from those of conventional linear PSt because of its unique 2D topology. In general, crosslinking reactions increase the $T_g$ of a bulk polymer compared to its pristine linear counterpart due to its 3D networking structure with fixed local and global chain mobilities[47,50,51]. However, we found that the $T_g$ of PSt-**1** is significantly lower ($T_g = 95$ °C) than that of the non-crosslinked linear PSt-L ($T_g = 103$ °C) synthesised

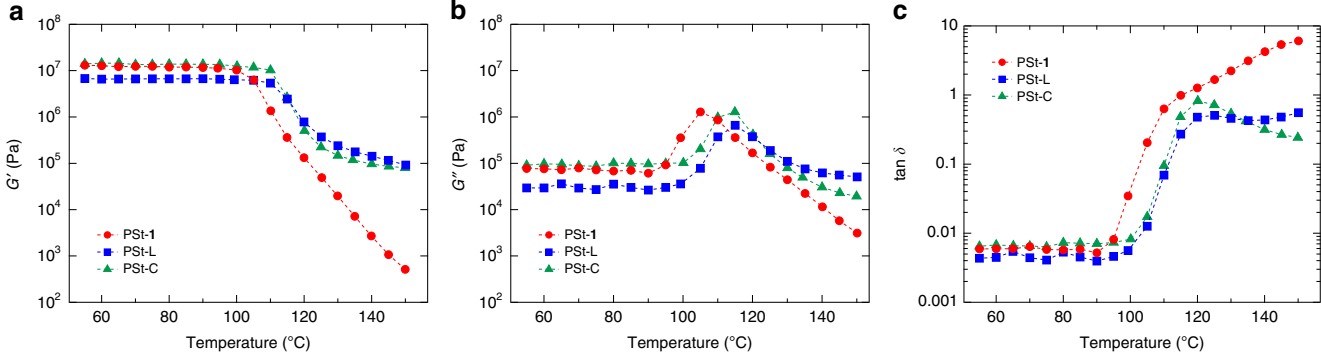

**Fig. 5 DMA data for PSt-1, PSt-L, and PSt-C.** Temperature sweeps of: **a** storage modulus ($G'$), **b** loss modulus ($G''$), and **c** loss factor (tan $\delta = G''/G'$) for PSt-**1** (red), PSt-L (blue), and PSt-C (green) at a frequency $\omega$ of $1\,s^{-1}$.

by solution-phase free-radical polymerisation. For further comparison purposes, we prepared a 3D-crosslinked (microgel) PSt, referred to as PSt-C, by the microemulsion polymerisation method (Supplementary Methods)[48]. PSt-C showed a higher $T_g$ of 107 °C as expected, meanwhile its DLS-based particle size (~150 nm diameter in chloroform) and crosslinking degree (~1%) are comparable to those of PSt-**1** (Supplementary Figs. 13 and 14). In the case of PSt-**1**, individual polymer sheets can adopt many possible conformations, with entanglements not permitted even at a very-high molecular weight of ~$3 \times 10^5$ g mol$^{-1}$, which is likely to engender the polymer with higher chain mobility on the molecular scale, resulting in a lower $T_g$ compared to PSt-C.

DMA revealed that PSt-**1** exhibits atypical mechanical behaviour. Figure 5 shows the temperature dependences of the storage ($G'$) and loss ($G''$) moduli of PSt-**1**, PSt-L, and PSt-C. DMA was carried out under oscillatory shearing at a frequency $\omega$ of $1\,s^{-1}$. Interestingly, PSt-**1** showed a continuous decrease in $G'$ above its $T_g$, and eventually exhibited flowing melt character, with a $G'$ four orders of magnitude lower than those of PSt-L and PSt-C at 150 °C (Fig. 5 and Supplementary Movie 1). This is also evidenced by the temperature dependence of the loss factor (tan $\delta = G''/G'$) of PSt-**1** that becomes higher than unity above its $T_g$, indicating that the material is more fluid-like than solid-like at this temperature and given frequency. On the other hand, PSt-L and PSt-C displayed typical viscoelastic behaviour commonly observed for bulk PSt. Master curves for these samples were constructed by shifting and superimposing multiple temperature–frequency data at a reference temperature $T_{ref}$ of 160 °C (Supplementary Fig. 15). PSt-L exhibited a rubbery plateau, as is commonly observed for ordinary PSt with a molecular weight higher than the critical entanglement molecular weight $M_c$ (typically 36 kg mol$^{-1}$ for PSt)[52]. In contrast, PSt-**1** did not reveal any rubbery plateau region in the master curve, irrespective of its high molecular weight, which supports our hypothesis that PSt-**1** is less inter-molecularly entangled due to its unique 2D network topology. The absence of such a rubbery plateau is rarely observed for a polymeric system with such a high $M_w$. Ring polymers and dendritic macromolecules are limited representatives that show fast relaxation processes and unentangled behaviour because of their unique terminal-free or spherical molecular configurations, respectively[53–56]. On the other hand, PSt-C showed a lack of chain relaxation in the high temperature region because of its fixed internal conformation with permanent crosslinks (Supplementary Fig. 15). These observations indicate that polymer 2D monosheets have very different MD and relaxation mechanism from conventional polymeric materials. Of note, PSt-**1** exhibited a large change in the magnitude of $G'$ over a small range of temperatures (~50 °C), which potentially offers high processability for common

manufacturing processes, including coating, hot-pressing, and the extrusion moulding of plastics. In addition, owing to its high solubility in organic media, PSt-**1** also allows solution-based processing. The film of PSt-**1** fabricated by solution casting method was rather brittle at room temperature because of suppressed chain entanglements, but showed smooth surface morphology on the SEM micrographs, indicating good processability due to the high dispersibility (Supplementary Fig. 16). These observations could show a useful aspect of the polymeric monosheets for their practical use.

## Discussion

We synthesised ultrathin polymeric 2D sheets of vinyl polymers by exploiting the nanospaces in pillared-layer MOFs as reaction platforms. Unlike conventional topochemical and exfoliation-based syntheses of polymeric nanosheets, the dimensionality and thickness of the nanosheet can be programmed by the structure of the host MOF. In addition, this 2D MOF-templated reaction system renders 2D sheets with enriched productivities and scalabilities that facilitate the determination of many properties that require large amounts of sample. The unimolecularly thick polymer 2D sheets exhibit exceptional thermal and mechanical behaviour that originate from their unique molecular topologies. The present study provides a universal method for the dimensional control of polymeric materials that have not been accessible to date by general solution and bulk-phase polymerisation processes.

## Methods

**Synthesis of MOF 1 and 2.** MOF **1** was prepared according to a literature procedure[45,46], while **2** was newly synthesised in a similar way to **1**. The synthesis of **2** is described in detail. A mixture of Ni(NO$_3$)$_2$·6H$_2$O (0.145 g), H$_3$btc (0.105 g), and dpb (0.116 g) in DMF (10 mL) was heated at 130 °C for 2 days. The product was washed with DMF and isolated by filtration to give green crystals of **2** with DMF as the guest solvent (0.34 g). For single-crystal X-ray diffractometry (SXRD), a single crystal of **2** was selected from the mother liquor and mounted in the single-crystal X-ray diffractometer. The crystal was then subjected to SXRD at –170 °C under a flow of nitrogen gas. Further details are provided in the Supplementary Methods.

**Crosslinking copolymerisation in the host frameworks.** Crosslinking polymerisation of St in **1** is described in detail. Dried host **1** (6.2 g) was prepared by heating at 130 °C for 12 h in vacuo (<0.2 kPa) in a Pyrex reaction tube. A mixture of St (20 mL), EDMA (32.1 mg), and AIBN (125 mg) was then added to the flask at room temperature. The mixture was left to stand for 30 min to allow the monomer and the initiator to diffuse into **1**, after which excess St was removed in vacuo (0.2 kPa) for 1.5 h at room temperature. The total amount of the above ingredients (St, EDMA, and AIBN) impregnated into **1** was 1.91 g, where all AIBN and EDMA added were incorporated into the pore. Full incorporation of AIBN and EDMA was confirmed by XRPD profile of the composite (**1**⊃St) that showed no diffraction peak originating from bulk AIBN and EDMA existing outside of **1** crystals. The reaction tube was then purged with nitrogen and a small portion of **1**⊃St was

removed from the tube and used to determine the actual amount of St and EDMA incorporated in **1** by NMR spectroscopy. Prior to NMR analysis, the composite was digested in a 1:9 (v/v) mixture of DCl/DMSO-$d_6$. The mixture was then subjected to $^1$H NMR spectroscopy (Supplementary Fig. 17). The reaction tube was purged and refilled with nitrogen, and then heated to 70 °C for 48 h for polymerisation. A small portion of the resultant composite was removed to determine the monomer conversion by NMR spectroscopy. The conversion of St was found to be 66%, while EDMA was completely consumed during the reaction. The composite prepared in this manner was washed with methanol to remove unreacted monomer inside the pores, and then stirred overnight in a 1:1 (v/v) mixture of 1 M aqueous HCl and methanol (200 mL) to decompose the host MOF. PSt-**1** was collected by filtration and then dried under reduced pressure at room temperature (1.1 g, 57% yield based on impregnated St). The $^1$H NMR spectrum of PSt-**1** was acquired in CDCl$_3$ as the solvent and showed typical broad peaks corresponding to the PSt backbone (Supplementary Fig. 2), while the Fourier-transform infrared spectrum of PSt-**1** showed a carbonyl-vibration peak at 1730 cm$^{-1}$ corresponding to the crosslinker (EDMA), confirming successful crosslinking and the formation of the networking architecture (Supplementary Fig. 18). An averaged crystallite size of the host framework, **1**, was estimated to be ~80 nm by means of peak-shape analysis on the XRPD data of **1**⊃PSt (Fig. 3a) using Scherrer's equation. PMMA-**1** was synthesised in identical fashions from MMA in 57% yield. Likewise, PSt-**2** was prepared in 25% yield using **2** as the host framework.

## Data availability

Crystallographic data for **2** have been deposited at the Cambridge Crystallographic Data Centre, under deposition number CCDC 1905001. A copy of the data can be obtained free of charge at www.ccdc.cam.ac.uk/data_request/cif. All other data supporting the findings of this study are available within the Article and its Supplementary Information, or from the corresponding author upon reasonable request.

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

## Acknowledgements

We thank Prof. K. Mayumi (The University of Tokyo) for technical support and useful DMA-analysis discussions. Profs. K. Ito (The University of Tokyo) and S. Kitagawa (Kyoto University) are acknowledged for providing access to DMA and AFM instrumentation, respectively. This work was supported by a JST-CREST programme (JPMJCR1321) and a Grant-in-Aid for Science Research on Innovative Area "Coordination Asymmetry" (JP16H06517) from the Ministry of Education, Culture, Sports, Science and Technology, Government of Japan.

## Author contributions

N.H., S.M., and Y.H. performed experiments associated with polymer synthesis, crystal growth, crystal structure determination, calorimetry, and mechanical tests. T.U. conceived the project and directed the research. All authors contributed to the writing and editing of the manuscript.

## Competing interests

The authors declare no competing interests.
