## [Peer Review File · Nature Communications]

REVIEWER COMMENTS

Reviewer #1 (Remarks to the Author):

The authors reply for the submitted manuscript "Unimolecularly thick monosheets of vinyl polymers fabricated in metal-organic frameworks by Uemura" and coworkers addresses my previous concerns. In my opinion the manuscript is suitable to be accepted in Nat. Commun.

There is only one question left that should be addressed. Why do the authors not include some additional comments and discussion including literature as back up from the replies into the manuscript? It seems that some points were raised by several reviewers so it would be favorable to include some additional text based on the replies to avoid misunderstandings for the reader.

Reviewer #2 (Remarks to the Author):

Dear Editor,

The authors reported a promising method to prepare 2D polymers in MOF cavities, which can be widely used to synthesize different 2D polymers. I still hope to see their SEM/TEM images as I recommended before. It is very helpful to demonstrate the morphology of their 2D polymer at free-standing state, sheets or shrink to be balls? Electrons may damage crystallinity of nano-sheets, but the main morphology may remain intact. For references, they over-cited papers from the same author. This is not good because it has the suspicion of guessing and apple-polishing reviewers. Anyway, I fully appreciate its novelty and it can be accepted after revisions following above suggestions.

Reviewer #3 (Remarks to the Author):

The manuscript is much easier to read and understand as in its previous form. Thanks to the authors for going through this effort. It was worth it.

There are a couple of remarkable aspects with the science reported:

1. The construction of a monolayer 2D network with irregular structure (absence of repeat units). This is a valuable, everything else but trivial addition to the field of synthetic 2D materials, in particular organic 2D materials
2. The facts that this network with its unprecedented topology consists of structural elements identical to those of the mass product polystyrene and that it can be obtained on a useful scale. This enables meaningful comparisons with linear and 3D cross-linked polystyrenes, which is the basis for finding out what the properties inherent to a macromolecule with 2D topology are like. There is practically no knowledge about what to expect from 2D networks, may they be regular (2D polymers) or irregular (as described here). Gaining this knowledge is thus of fundamental importance. Having even thin stacks of sheets of some 2D material does not help. There is no way around monolayers in large quantity. This is what the paper offers.
3. The finding that the glass transition of the 2D material created is lower than that of reference systems and the interpretation that this may be due to the inherent lack of entanglements. This relates directly to point 2 and will be of great interest to all those, interested in the viscoelastic behaviour of polymeric materials. This observation also urges the question whether the reported T_g

difference to the linear material is real. While there is no doubt that the T_g of the linear polymer is in the regime of molar mass independence, it is simply unknown whether this is also so for a 2D network of the same molar mass. For those the property-molar mass relation may be different. Already the fact that this question arises testifies the interest the reported structures will find.

These three aspects have considerable weight and make clear why this work principally deserves being published in Nature Communications.

Nevertheless, even in the latest version the reviewer feels that the authors could do a bit more in terms of describing the network (beyond just the molar mass) and of quantifying more rigorously the cross-link density. It would be handy to have an estimate of the the number of cross-links per unit area, the average segment length between the cross-links and the estimated lateral extension of the unfolded monolayer network. A small graphic showing these features would also be nice (even in the main text).

Concerning the experimental proof for cross-linking density, the authors should please explain the role of the very low intensity signals in the ¹H NMR spectrum of their network appearing around 5 ppm. Even though at first glance one may not realize them, they are there. There should be an insert with this region being amplified and, if possible, a signal assignment. It almost looks like as if the cross-linker had not reacted completely, which may not be a surprise given the narrow confinement. If this is so, there will be some dangling acrylate units. They will need to be considered in the structural model requested. This would also change the proposed cross-linking degree of 1%.

Do the authors have an explanation for why the TEM images in Figure S2 are around 50 μm in size, while the sizes of the features in the AFM images are much smaller. Shouldn't they ideally have the same size? Please provide a comment.

Since the whole concept depends on the quality of the MOF mother crystals, it appears appropriate to make a few statements on how good these crystals are. Why not showing a reciprocal space reconstruction in the SI. Are there no signs of mosaicity? They could question the concept. The authors must have all this. Why not showing it?

In sum, this is a truly excellent paper presenting an exciting new material, which should be published after some improvements.

Point-to-Point Responses to Referee's Comments

We uploaded a copy of the revised manuscript tracking all changes in blue color as a Review-Only Material.

For Reviewer 1:

The authors reply for the submitted manuscript "Unimolecularly thick monosheets of vinyl polymers fabricated in metal-organic frameworks by Uemura" and coworkers addresses my previous concerns. In my opinion the manuscript is suitable to be accepted in Nat. Commun.

There is only one question left that should be addressed. Why do the authors not include some additional comments and discussion including literature as back up from the replies into the manuscript? It seems that some points were raised by several reviewers so it would be favorable to include some additional text based on the replies to avoid misunderstandings for the reader.

=>We thank this reviewer for suggestions. In the revised manuscript, we added discussions that reflect the points raised by previous/present reviewers. The discussions added to the revised manuscript and Supplementary Information are summarized below.

1. **SEM Image.** As per the suggestion from Reviewer 2, SEM images of the PSt-1 cast films were taken and added in Supplementary Figure 16. Please refer to the answer to Reviewer 2's comments for details.
2. **AFM Analysis.** We added the AFM analysis data for molecular weight calculation in Supplementary Figure 11, which is in response to the comment from previous Reviewer 2. We carried out statistical analysis on the AFM image of PSt-1 to calculate statistical molecular weights of PSt-1 monosheets. The area of each monosheet (particle) is determined by particle analysis (masking with height threshold) on the AFM image. PSt-1 was deposited on HOPG substrate at which the monosheets adopt expanded conformation. Based on the area (A), sheet height ($H = 0.7$ nm), and density of polystyrene (1.04 g/mol), the molecular weight of each individual monosheet is calculated by the equation given in the figure. Molecular weight, M_n , is determined to be 5.6×10^6 g/mol by this statistical analysis ($N = 147$). Although the AFM-based value is larger than the SLS-based value (298,000 g/mol), this can be explained by the difference in the way on analyses. Because of the imaging resolution limit, AFM is not able to visualize low-molecular weight fraction of PSt-1, which results in the overestimated M_n . In addition, the monosheets should have lower density compared with bulk PSt because of loose molecular packing (W. J. Orts *et al.*, *Phys. Rev. Lett.* **1993**, *71*, 867.) due to the unimolecular thickness and potential porosity in the mesh structure (Supplementary Figure 6). Therefore, both AFM- and SLS-based values could be in the range of the molecular weight distribution.

$$A \times H \times D_{\text{PSt}} \times N_A = MW$$

A : Masked area
 H : Sheet height (0.7 nm)
 D_{PSt} : Density of polystyrene (1.04 g/cm³)
 N_A : Avogadro's number
 MW : Molecular weight of a sheet

Calculated M_n from AFM image
 ($N = 147$)
 5.3×10^6 g/mol

[Added] Supplementary Figure 11. The molecular weight of PSt-1 was calculated by means of particle analysis on AFM data. (a) An AFM topographic image of dispersed monosheets of PSt-1 deposited on HOPG surface. PSt-1 monosheets with a wide variety of sizes (ca. 50-300 nm) are observed. (b) Particle analysis is applied to the AFM image of panel a. The area of each PSt-1 sheet is measured for $N = 127$ sheets and statistically analyzed to calculate the averaged molecular weight based on the given equation in the figure. The density of polystyrene of 1.04 g/cm³ was used. Although the AFM-based value is larger than the SLS-based value (298,000 g/mol), this can be explained by the difference in the way on analyses and lower density of polystyrene. Because of the imaging resolution limit, AFM is not able to visualize low-molecular weight fraction of PSt-1, which results in the overestimated M_n . In addition, the monosheets should have lower density compared with bulk polystyrene because of loose molecular packing due to the unimolecular thickness⁵ and potential porosity in the mesh structure (Supplementary Figure 6 and 12). Therefore, both AFM- and SLS-based values could be in the range of the molecular weight distribution.

- 3. Network Structure.** We added discussions and a schematic picture for the plausible structure of PSt-1 monosheet in Supplementary Figure 6, which are in response to Reviewer 3's comments and concerns. Please refer to the answer to Reviewer 3's comments for details.
- 4. Porosimetry Data.** We added the results of porosimetry measurements that give an insight of material morphology in Supplementary Figure 12 of the revised manuscript. We consider that it is useful to avoid misunderstandings for the reader. As seen in the AFM image of Supplementary Figure 6, the PSt-1 monosheet has many transient pores. This morphology may be formed through the transcription of the host MOF structure. The pore size distribution was calculated to be ~20 nm by NLDFT simulation method, which could reflect the film morphology at dried state. Since the mesh structure should be collapsed and shrunk at the gas adsorption measurement condition, the mesoporous nature observed in this measurement is also caused by voids in the agglomerated

structure of polymer particles.

[Added] Supplementary Figure 12. (a) N₂ adsorption isotherm of PSt-1 at 77 K, (closed symbol) adsorption curve; (open symbol) desorption curve. The isotherm shows typical behavior of mesoporous material. (b) Pore-size distribution of PSt-1 calculated using NLDFT method. W denotes pore width (nm). The distribution has a peak at $W = 19.5$ nm. The porosimetry results indicate mesoporous nature of PSt-1 at dried state. The calculated pore size is ~ 20 nm, which could reflect the film morphology. Since the mesh structure can be collapsed and shrunk at the dried condition, the observed mesoporous nature could be also attributed to the agglomerated morphologies of the polymer particles.

5. **NMR Analysis.** As per suggestion and comments from Reviewer 3 in the previous round of review, we examined again the NMR data of PSt-1 and made discussions considering unreacted (dangling) crosslinkers. The corresponding figure (Supplementary Figure 2) was revised and detailed discussions are given in the caption. Please refer to the answer to Reviewer 3's comments for details.
6. **Crystal Size.** Discussions about the discrepancy between the sizes of MOF crystal and monosheet are given in the main text (page 9, line 8-17), which is in response to Reviewer 3's comments for details. Please refer to the answer to Reviewer 3's comments for details.

For Reviewer 2:

The authors reported a promising method to prepare 2D polymers in MOF cavities, which can be widely used to synthesize different 2D polymers. I still hope to see their SEM/TEM images as I recommended before. It is very helpful to demonstrate the morphology of their 2D polymer at free-standing state, sheets or shrink to be balls? Electrons may damage crystallinity of nano-sheets, but the main morphology may remain intact. For references, they over-cited papers from the same author. This is not good because it has the suspicion of guessing and apple-polishing reviewers. Anyway, I fully appreciate its novelty and it can be accepted after revisions following above suggestions.

=>We appreciate this reviewer's suggestion. We added SEM images of the cast film of PSt-1 in Supplementary Figure 16. The film of PSt-1 was prepared by casting from its chloroform solution (40 mg/mL) on a glass substrate and coated with Pt before observation. The film was rather brittle and glassy at room temperature. It shows typical smooth surface morphology with sharp edges of cleavage, indicating good processability of the monosheets.

[Added] Supplementary Figure 16. SEM micrographs of the PSt-1 cast film. The film (thickness: ~3 μm) was prepared by casting from its chloroform solution (40 mg/mL) on a glass substrate.

=>We also added comments on the SEM images and a property of the PSt-1 film in the main text as follows.

[Added] (page 11, line 4-9) “Additionally, owing to its high solubility in organic media, PSt-1 also allows solution-based processing. The film of PSt-1 fabricated by solution casting method was rather brittle at room temperature because of suppressed chain entanglements, but showed smooth surface morphology on the SEM micrographs, indicating good processability due to the high dispersibility (Supplementary Figure 16). These observations could show a useful aspect of the polymeric monosheets for their practical use.”

=>We reconsidered the literature citation and removed/added following papers.

[Removed] in order to avoid multiple citations for the same topic.

Ref. 11. Bholá, R. *et al.* A two-dimensional polymer from the anthracene dimer and triptycene motifs. *J. Am. Chem. Soc.* **135**, 14134–14141 (2013).

Ref. 16. Bauer, T. *et al.* Synthesis of free-standing, monolayered organometallic sheets at the air/water interface. *Angew. Chem. Int. Ed.* **50**, 7879–7884 (2011).

Ref. 19. Feng, X. & Schlüter, A. D. Towards macroscopic crystalline 2D polymers. *Angew. Chem. Int. Ed.* **57**, 13748–13763 (2018).

Ref. 34. Kunitake, T. Synthesis of ultrathin polymer films by self-assembly. *Macromolecular Symposia* **98**, 45–51 (1995).

Ref. 37. Blumstein, A. & Ries, H. E. Monolayer properties of a crosslinked insertion poly(methyl methacrylate). *J. Polym. Sci. B* **3**, 927–931 (1965).

[Added] in terms of recent progress of MOF-templated polymerization technique.

New Ref. 42. Lee, H.-C. *et al.* Toward Ultimate Control of Radical Polymerization: Functionalized Metal–Organic Frameworks as a Robust Environment for Metal-Catalyzed Polymerizations. *Chem. Mater.* **30**, 2983–2994 (2018).

For Reviewer 3:

The manuscript is much easier to read and understand as in its previous form. Thanks to the authors for going through this effort. It was worth it.

There are a couple of remarkable aspects with the science reported:

1. The construction of a monolayer 2D network with irregular structure (absence of repeat units). This is a valuable, everything else but trivial addition to the field of synthetic 2D materials, in particular organic 2D materials
2. The facts that this network with its unprecedented topology consists of structural elements identical to those of the mass product polystyrene and that it can be obtained on a useful scale. This enables meaningful comparisons with linear and 3D cross-linked polystyrenes, which is the basis for finding out what the properties inherent to a macromolecule with 2D topology are like. There is practically no knowledge about what to expect from 2D networks, may they be regular (2D polymers) or irregular (as described here). Gaining this knowledge is thus of fundamental importance. Having even thin stacks of sheets of some 2D material does not help. There is no way around monolayers in large quantity. This is what the paper offers.
3. The finding that the glass transition of the 2D material created is lower than that of reference systems and the interpretation that this may be due to the inherent lack of entanglements. This relates directly to point 2 and will be of great interest to all those, interested in the viscoelastic behaviour of polymeric materials. This observation also urges the question whether the reported T_g difference to the linear material is real. While there is no doubt that the T_g of the linear polymer is in the regime of molar mass independence, it is simply unknown whether this is also so for a 2D network of the same molar mass. For those the property-molar mass relation may be different. Already the fact that this question arises testifies the interest the reported structures will find.

These three aspects have considerable weight and make clear why this work principally deserves being published in Nature Communications.

1. Nevertheless, even in the latest version the reviewer feels that the authors could do a bit more in terms of describing the network (beyond just the molar mass) and of quantifying more rigorously the cross-link density. It would be handy to have an estimate of the number of cross-links per unit area, the average segment length between the cross-links and the estimated lateral extension of the unfolded monolayer network. A small graphic showing these features would also be nice (even in the main text).

⇒ In the newly added Supplementary Figure 6 (given below), we show the AFM image of PSt-1 and a schematic picture of the plausible network structure of PSt-1 with a crosslinking density of 1 %. The AFM image newly taken at milder condition (softer AFM cantilever) shows knit-like mesh structure

of many small transient holes (panel a), which is consistent with the envisaged appearance based on the calculated data (panel c).

[Added] Supplementary Figure 6. (a) An AFM topographic image of PSSt-1 deposited on HOPG and (b) the height profile along the white line in panel a. The cantilever, OMCL-AC200TSA (Olympus), with spring constant of 9 pN/nm and 150 kHz resonant frequency was used, which gives a milder condition than that is used for the AFM imaging of Figure 4. (c) A schematic illustration of speculated network structure of each PSSt-1 monosheet. The numbers of crosslinks and molecular weight values are calculated based on the experimental data determined by NMR and SLS analysis and a hypothesis that each chain has one AIBN end group.

=>In addition to Supplementary Figure 6, we added the discussion regarding this analysis in the main text as follows.

[Added] (page 7, line 20-23) “The network structure of individual PSSt-1 monosheet could be roughly estimated by numerical calculation based on NMR data and the crosslinking density. Each PSSt-1 monosheet has ~30 crosslinking points inter-connected with polystyrene chains with the molecular weight of ~5 kg/mol, forming the mesh-like 2D networking structure (Supplementary Figure 6).”

2. Concerning the experimental proof for cross-linking density, the authors should please explain the role of the very low intensity signals in the ¹H NMR spectrum of their network appearing around 5 ppm. Even though at first glance one may not realize them, they are there. There should be an insert with this region being amplified and, if possible, a signal assignment. It almost looks like as if the cross-linker had not reacted completely, which may not be a surprise given the narrow confinement. If this is so, there will be some dangling acrylate units. They will need to be considered in the structural model requested. This would also change the proposed cross-linking degree of 1%.

=>We examined the NMR data as described in the previous response letter. In the current manuscript, we added the NMR spectrum with the magnified views (Supplementary Figure 2). When taking into account the small proton peaks derived from unreacted EDMA, the crosslinking density is 0.85%. We consider that this value is in the range of experimental/analytical error, thus we use the rounded value, ~1%, in the manuscript. Detailed calculations and explanations are given in the caption of Supplementary Figure 2.

[Replaced] Supplementary Figure 2. ^1H NMR spectrum (CDCl_3) of isolated PSt-1 measured at 25 °C. No proton signal originating from the residual ligands (Hbtc and bpy) was observed, which supports the successful removal of MOF host. In addition, small signals of end-groups derived from AIBN were observed in 1.1–0.9 ppm. Based on area integration analysis using the end groups and main chain peaks, an averaged degree of polymerization (DP) is calculated to be ~ 800 ($M_n = \sim 83,000$) when assuming linear chains without crosslinking. Here we made the assumption that all polystyrene chains are end-capped with the AIBN fragment at one end. Considering the absolute molecular weight of PSt-1 ($M_w = 298,000$), this calculation provides us a useful aspect of molecular structure of PSt-1 in which 3.6 linear chains with an average molecular weight of $\sim 83,000$ are crosslinked each other. A schematic illustration of plausible network structure is given in Supplementary Figure 6.

Insets show the magnified spectra of the proton signals in 5–6 ppm. Small peaks at 5.34 and 5.54 ppm can be attributed to the protons of unreacted methacrylate unit of EDMA (2H, integration area: A). The methylene protons of EDMA are observed at 3.5 ppm as a broad peak (4H, integration area: B). The ratio of those integration values ($2A/B$) becomes ~ 0.15 , which indicates that the crosslinking degree is $0.85\% = \sim 1\%$.

3. Do the authors have an explanation for why the TEM images in Figure S2 are around 50 μm in size, while the sizes of the features in the AFM images are much smaller. Shouldn't they ideally have the same size? Please provide a comment.

=>Seemingly, a micrometer-sized MOF crystal would give very long polymers with the molecular weight more than $> 10^6$ according to their size scale. However, this is unlikely in the practical case because AIBN initiates the polymerization inside the MOFs to form many polymer domains randomly in the MOF crystals. Thus, molecular weight of resulting polymers is usually 10,000 ~ 500,000 although morphology of the polymer assemblies is macroscopically retained after the removal of MOF templates (*Chem. Mater.* **2013**, *25*, 3772). In response to this question from the reviewer, we added the detailed comments in the main text as follows.

[Added] (page 9, line 8-17) "It should be noted that the size of polymer monosheets obtained was generally much smaller (~ 100 nm) than the size of host MOF crystals (~ 50 μm). This is typically observed in the confined polymerisation system using MOFs. AIBN loaded in the MOF generates free radicals to start polymerisation at random locations in the crystals. Volume shrinkage during polymerisation (e.g. volume of dimer $<$ volume of two monomers) results in the formation of many polymer domains isolated in the MOF crystals. In addition, diffusion of the polymer domains is highly restricted because of the confined feature of this reaction system. This situation leads to depletion of monomers, which ends up with smaller size of product compared to the size of host MOF crystals. As also evidenced by the AFM-based analysis of PSt-1, which showed size distribution of the monosheets (Supplementary Figure 11), the size control is the next challenge for this system."

4. Since the whole concept depends on the quality of the MOF mother crystals, it appears appropriate to make a few statements on how good these crystals are. Why not showing a reciprocal space reconstruction in the SI. Are there no signs of mosaicity? They could question the concept. The authors must have all this. Why not showing it?

=>To load the monomers in the MOF template, the as-synthesized MOF crystals have to be dried and evacuated the included solvent (DMF) under a dynamic vacuum at 120 $^{\circ}\text{C}$. This evacuation process causes many cracks in the single crystal of MOF, which leads to the loss of single crystallinity. Most single crystals break up into powdery small crystals which are no longer analyzable by single crystal X-ray diffraction (SXRD) method. Due to the above reasons, we were not able to measure actual mosaicity and reciprocal space mapping data for the single crystal of **1**. Nonetheless, the powdery crystals of **1** shows satisfactory diffraction pattern on the powder X-ray diffraction (PXRD) measurements (Figure 3a). The PXRD data ensures that the framework structure is preserved even after the evacuation process. In addition, the sharp peak shape indicates that the sample is highly crystalline even in the powdery form.

=>Instead of mosaicity, there is another practicable method to quantify the crystal quality, that is crystallite size. We estimated the crystallite size of **1** by means of peak-shape analysis on the PXRD data (Figure 3a) using Scherrer's equation. The averaged crystallite size of **1** was calculated to be 80 nm, which is indicative of a high degree of crystallinity often observed for common MOFs.

=>From our experiences in MOF-templated polymerizations, we are aware that the molecular weight of the synthesized polymer is not always dependent on the crystal quality. A highly crystalline MOF with a large crystallite size does not necessarily give very high molecular weight polymers. Therefore, we do not think that the mosaicity is crucial in the present system and rather think that the polymerization proceeds beyond the domain boundaries of crystallite (the size of crystallites was smaller than the monosheet size determined by DLS.). However, we should also note that low-quality MOF crystals are not suitable for the MOF-templated polymerization, indeed. When we used such low-quality crystals (i.e. small crystallite size/high mosaicity), the polymerization in MOFs often failed in our experiences. In this case, such a high degree of mosaicity would be the dominant cause of the problem.

=>To ensure the reproducibility of experiments, we added a note about the quality of the MOF crystals in Methods section as follows.

[Added] (page 13, line 3-5) “An averaged crystallite size of the host framework, **1**, was estimated to be ~80 nm by means of peak-shape analysis on the PXRD data of **1**⊃PSt (Figure 3a) using Scherrer’s equation.”

In sum, this is a truly excellent paper presenting an exciting new material, which should be published after some improvements.

=>We appreciate this reviewer’s comments and suggestions that are very useful to improve our work and deepen the discussions.

REVIEWERS' COMMENTS:

Reviewer #1 (Remarks to the Author):

The revised version of the manuscript "Unimolecularly thick monosheets of vinyl polymers fabricated in metal-organic frameworks" by Uemura and coworkers addresses my final comment perfectly. In my opinion it can be accepted in Nat. Commun. in the present version.

Reviewer #2 (Remarks to the Author):

Dear Editor,
I fully appreciate this work and it can be accepted now.

Reviewer #3 (Remarks to the Author):

The authors did an excellent job in addressing the concerns of the reviewers. Reviewer 3 particularly likes the changes regarding the structural aspects of the network reported which culminate in the new Supplementary Figure 6 and the additional lines 20-23 on pg. 7 in the main text. This reviewer also accepts the comments concerning crystal quality and NMR analysis.

The reported science has substantial potential and is likely to give innovative, non-traditional polymer chemistry a strong push. Congratulations.

The manuscript has now reached a matureness that publication can be recommended as it stands.

Point-to-Point Responses to the Reviewer's Comments

Reviewer #1 (Remarks to the Author):

The revised version of the manuscript "Unimolecularly thick monosheets of vinyl polymers fabricated in metal–organic frameworks" by Uemura and coworkers addresses my final comment perfectly. In my opinion it can be accepted in Nat. Commun. in the present version.

Reviewer #2 (Remarks to the Author):

Dear Editor,

I fully appreciate this work and it can be accepted now.

Reviewer #3 (Remarks to the Author):

The authors did an excellent job in addressing the concerns of the reviewers. Reviewer 3 particularly likes the changes regarding the structural aspects of the network reported which culminate in the new Supplementary Figure 6 and the additional lines 20-23 on pg. 7 in the main text. This reviewer also accepts the comments concerning crystal quality and NMR analysis.

The reported science has substantial potential and is likely to give innovative, non-traditional polymer chemistry a strong push. Congratulations.

The manuscript has now reached a matureness that publication can be recommended as it stands.

=>We would like to thank all reviewers for their helpful and expert comments. The suggestions from those reviewers were very useful to improve the quality of our work and deepen the discussions.